# Experiences of coping with the first wave of COVID-19 epidemic in Philadelphia, PA: Mixed methods analysis of a cross-sectional survey of worries and symptoms of mood disorders

**Igor Burstyn**[ID]◯*, **Tran B. Huynh**◯

Department of Environmental and Occupational Health, Dornsife School of Public Health, Drexel University, Philadelphia, Pennsylvania, United States of America

◯ These authors contributed equally to this work.
* igor.burstyn@drexel.edu

## Abstract

Our objective was to describe how residents of Philadelphia, Pennsylvania, coped psychologically with the first wave of COVID-19 pandemic. In a cross-sectional design, we aimed to estimate the rates and correlates of anxiety and depression, examine how specific worries correlated with general anxiety and depression, and synthesize themes of "the most difficult experiences" shared by the respondents. We collected data through an on-line survey in a convenience sample of 1,293 adult residents of Philadelphia, PA between April 17 and July 3, 2020, inquiring about symptoms of anxiety and depression (via the Hospital Anxiety and Depression Scale), specific worries, open-ended narratives of "the most difficult experiences" (coded into themes), demographics, perceived sources of support, and general health. Anxiety was evident among 30 to 40% of participants and depression—about 10%. Factor analysis revealed two distinct, yet inter-related clusters of specific worries related to mood disorders: concern about "hardships" and "fear of infection". Regression analyses revealed that anxiety, depression, and fear of infection, but not concern about hardships, worsened over the course of the epidemic. "The most difficult experiences" characterized by loss of income, poor health of self or others, uncertainty, death of a relative or a friend, and struggle accessing food were each associated with some of the measures of worries and mood disorders. Respondents who believed they could rely on support of close personal network fared better psychologically than those who reported relying primarily on government and social services organizations. Thematic analysis revealed complex perceptions of the pandemic by the participants, giving clues to both positive and negative experiences that may have affected how they coped. Despite concerns about external validity, our observations are concordant with emerging evidence of psychological toll of the COVID-19 pandemic and measures employed to mitigate risk of infection.

**Data Availability Statement:** Data cannot be made publicly available due to potentially identifying information that the authors are not allowed to share per their IRB approval protocol. Please

contact the IRB at (267) 359-2471 or
HRPP@drexel.edu if you have any questions.
Professor H. Quick (https://drexel.edu/dornsife/
academics/faculty/Harrison%20Quick/; email:
hsq23@drexel.edu) kindly agreed to be non-author
contact on matters of data sharing.

**Funding:** The authors received no specific funding
for this work.

**Competing interests:** The authors have declared
that no competing interests exist.

## Introduction

Evidence of excess of anxiety and depression during the first wave of the coronavirus disease
(COVID-19) pandemic in the United States (US) was noticeable as early as April-June 2020 [1,
2]. The evidence of the elevated risk of conditions appeared to cluster among healthcare and
emergency workers who were likely to come into contact or anticipated contact with persons
ill with COVID-19 [1]. The Household Pulse Survey by US Centre for Disease Control and
Prevention (US CDC) that started in April 2020 reported, for period ending March 1, 2021, the
overall rates of anxiety and depression disorders in the order of 40%; elevated rates in younger
ages, among women, and with lower education; reduced rates among non-Hispanic White and
Asian races/ethnicities were noted [3]. The observed rates were 3 to 4 times higher than for the
same periods previous year [3–5]. Fitzpatrick et al. [6] surveyed a representative sample of
10,368 adults in the US in late March 2020, regarding general fear and anxiety related to
COVID-19, social and behavioural health changes, health, and demographics. They reported
more than a third of respondents with symptoms of depression, higher among women, His-
panic, unmarried, unemployed, fearful of infection, with food insecurity, pessimistic, and those
who reported less control over their fate and weaker social ties [6]. Examination of mortality
statistics by Mulligan [7] reveals excess of "deaths of despair" (involving drug overdose, suicide,
or alcohol abuse) in 2020 (up to early October) in the US in age groups that do not have excess
risk of death from COVID-19: working age men. The estimated excess was of about 30,000
additional deaths above 66,000 expected for the entire year; roughly half of the excess was in
men of working age. The trends in these excess deaths track unemployment insurance claims;
it peaked in June and then levelled-off, it is consistent with increase in rates of mental ill-health
over the same period. The trends were similar yet less pronounced for women of working age.

The first case of COVID-19 in Philadelphia was announced on March 10, 2020, followed by
the stay-at-home order on March 23 [5, 8], which was relaxed on June 5 by transition from
"red" (most restrictive) to the "yellow" (less restrictive) phase of the Pennsylvania's reopening
plan. We reported that anxiety and depression in general population of Philadelphia during
the first wave of the pandemic can be due to widespread disruption of working lives, especially
in "non-essential" low-income industries, on par with experience in healthcare [9]. We verified
that levels of anxiety and depression were indeed on par with healthcare setting through a sur-
vey of physicians and nurses that gathered data using analogous instruments at the same time
in the same region [10]. We now turn our attention to broader range of experiences of persons
in our sample that transcend work-related circumstances.

One of the limitations of *academic* work published to date on the issue of mental health
impact of COVID-19 pandemic is that experiences of individuals coping with the events are
reduced to highly abstracted constructs and specific voices and nuanced experiences are thus
lost. We believe that understanding of how events were perceived and what the affected people
thought about these experiences, with focus on "the most difficult" events, is essential lest the
pandemic is de-humanised, and opportunities to understand both the breadth of its impact
and opportunities to help us all survive such events would be lost behind "the pandemic of
dashboards" (which capture next to nothing of the importance to either most people's daily
lives or risk assessment [11, 12]).

Our aim was to describe experiences during the first wave of COVID-19 pandemic that
affected mood disorders and pandemic-specific worries of residents of Philadelphia, PA,
regardless of whether they were employed during that time. Such descriptions are meant to
suggest aspects of lived experiences that may be considered for both monitoring of responses
to such events and possible opportunities to alleviate some aspects of psychological impact of
pandemics.

## Materials and methods

### Data collection

Data collection was previously described in our article focused on impact of changes to work on anxiety and depression among study participants [9]; the details are described below with emphasis placed on the aims of current analysis and its approach. The study was approved by the Institutional Review Board of Drexel University. Written consent was provided on the first page of the online survey. Only those who accepted the consent can proceed with the survey.

### Survey instrument

The survey contained blocks of questions that covered general health, demographics, worries and support, measures reported to mitigate risk, employment, COVID-19 testing, and Hospital Anxiety and Depression Scale (HADS) [13, 14]. HADS contains sub-scales that measure anxiety and depression separately, each ranging from 0–21, with scores ≥11 used to identify cases in the general adult population. HADS is accepted as valid for use in the general non-hospitalized population [15]. Two general perceived health questions from the SF-36 (36-Item Short Form Health Survey) were asked: (a) "in general, would you say your health is: excellent, very good, good, fair, poor" and "compared to other persons your age, would you say your health is: excellent, very good, good, fair, poor" [16]. Survey included a battery of questions about perceptions (captured on Likert-like scale ranging from 0 to 100) of source of anticipated support during pandemic and specific worries. "Worrying" is an established proximal antecedent of generalized anxiety (such as assessed by HADS) as opposed to a more distal "environmental" cause [17, 18]. Consequently, we did not adjust for worries in regression models of HADS scores described below, but rather investigated association between worries and HADS for anxiety in factor analysis and sought to identify latent epidemic-specific worries that were not captured by generalized anxiety and depression (details below in description of statistical analyses). We also asked participants to share their most difficult experiences during the epidemic and any other stories that they wished to share with us; treatment of these data is described below in thematic analysis section. The key questions we asked that are not present in the cited literature are reported as part of results below. The participants could choose to complete the survey in English, Spanish, Vietnamese, or Chinese.

### Recruitment

Eligible participants were adults aged 18 years and older and were living in Philadelphia, PA. The study was restricted to Philadelphia residents because the lockdown date and policies mitigating the pandemic differed by county. Our data collection started on April 17 and ended July 3, 2020, spanning both red and yellow phases of restrictions.

The online survey was administered via Qualtrics software (Qualtrics, Provo, UT). Participants were recruited using a convenience sampling approach via multiple communications strategies. Emails were sent to the investigators' network (heavily weighted to academic community), 623 registered community organizations using a publicly available roster (three mailings, between April 21 and May 13), and other community groups found on the internet. The survey was advertised in a neighbourhood online newspaper *West Philly Local* (on April 18) [19] and a regional newspaper's website the *Philadelphia Inquirer* (May 11–17). Starting May 19, we used Facebook to design and customize an advertisement campaign to place ads on its site and affiliated social media platforms (Instagram, Messenger, and the Facebook Audience Network), following methodology in Ali et al. [20].

## Quantitative analysis

There were missing values in most categorical and continuous variables in the total dataset of 1,577 responses. HADS scores with less than half of missing values were imputed with the individual subscale mean score [21]. Missing values of categorical variables were kept as is to stabilize regression analyses and more fully utilize the data. However, 284 respondents with missing information on either gender, employment, or those with more than half of responses to HADS scale were removed, leaving 1,293 for analysis.

Data was prepared for analysis in R [22]. All statistical calculations were performed in SAS v 9.4 (SAS Institute, Cary, NC). Association of HADS scores for anxiety and depression were examined for each of the covariate of interest in terms of counts of cases with scores $\geq 11$, means and standard deviations for categorical covariates and correlations for continuous covariates.

Multivariable regression models of HADS scores were estimated using negative binomial regression, suited for count data such as HADS scores. These yielded relative rates (RR) and 95% confidence intervals (CI) of change in HADS scores in relation to variables that showed evidence of association with HADS scores in univariate analyses.

Primary outcomes were HADS scores for anxiety and depression. In addition, we were interested in whether there was a clustering of "worries", known to be on the causal pathway towards mood disorders [17, 18], that indicate latent sub-clinical consequences of coping with the pandemic. To investigate this possibility, we conducted factor analysis on HADS scores plus responses to the "worries" questions. Orthogonal principal component rotation indicated that, as in our prior work in sub-sample of employed participants prior to the epidemic [9], all worries were related to mood disorders. Oblique VARIMAX rotation was therefore performed (the type of rotation did not alter the results) and factors extracted using scree plots and $\chi^2$ test. Three factors were expected *a prior* based on the content of the questionnaires. Factor loadings $\geq 0.3$ were interpreted and displayed in path diagram; all such loadings were confirmed in the maximum likelihood factor analysis to have $p < 0.001$ for a null hypothesis that they are equal to 0. Factors scores were generated and those not directly related to HADS scores were considered as outcomes in linear regression analyses due to their Gaussian distribution resulting from the additive procedure used to calculate factor scores (confirmed empirically).

The predictor variables considered in all regression analyses were the themes of most difficult experiences during the epidemic derived from thematic analysis of narrative responses (coded as present or absent, see below), employment status during epidemic (not employed at the onset, employed throughout, lost job during epidemic), number of days at data collection since start of epidemic, and perceived sources of support (see above), while controlling for health (general and recent malaise, reported positive test of COVID-19) plus demographics. We attempted to limit regression analyses to conditions that existed at the start of epidemic or were time-invariant during its course, as well as experiences that occurred during epidemic that were unlikely to be consequences of anxiety, depression or worries (e.g., measures taken to mitigate risk of infection were described but not seen as plausible antecedents of worries because we lacked information on time course of their occurrence). Because of known higher rates of mood disorders among women in general [23] and suspected greater impact of the pandemic on mental health of men [7], we estimated cross-product interactions of coded themes of most difficult experiences by gender; we interpreted only those interaction terms that had $p < 0.05$; furthermore, because of low counts of "other" gender, only effect estimates for males and females were interpreted. We thus undertook regression analyses mostly to succinctly describe experiences of our participants and are hesitant to draw causal inferences.

### Thematic analysis

The two investigators independently examined free text narratives supplied by the respondents in response to the following questions:

> "Considering events affecting your work during the COVID-19 epidemic (since 10th March) what has been the most difficult or stressful event you have had to deal with?"

> "Is there anything you would like to share with us?"

The investigators independently extracted themes and reconciled both the themes and classification of responses by consensus. There was little disagreement between the investigators in these matters. The fourteen themes were: change in work: self or family, change in schooling, childcare, lost income: self or family, unavoidable proximity to strangers/fear infection, social isolation, return to work fear (not analysed further here because we dealt with this issue though a separate variable in prior paper [9]), poor health: self or others, trouble sleeping, uncertainty, media coverage, food access, death: family or friend, and anger. Each response was coded as either belonging to the theme or not for regression analyses (detailed above); narrative summary of cross-cutting themes is also presented lest the richness of the stories shared by the respondents be lost in the abstraction of themes. Biserial correlation was used in relating HADS (used as a continuous score) to themes that were coded as present or absent. The pivotal *disclaimer* that must have affected thematic analysis is that both authors are essential workers and spouses of the same who lost childcare and had to work from home. They resided in Philadelphia since start of the epidemic. IB was annoyed by poor quality of data on the epidemic, lack of preparedness, and absence of transparency on reasons for policies related to epidemic.

## Results

### Description of sample

Table 1 presents description of the participants in our sample in terms of demographics and HADS scores. The patterns are not materially different for the minority (n = 369) who did not have a job at the start of the pandemic and their statistics are provided in S1 Table. The study participants were predominantly female, white, older than 35 years of age, with pre-pandemic personal income above $40,000 per year, college-educated, married or living as married, and having no children. Anxiety and depression scores were lower among males, African Americans, persons 55 years of age or older, with pre-pandemic personal income $100,000 per year or more, widowed or divorced, and those with no children < 18 years of age living at home; there was no material difference by education. The highest rates of anxiety (50% cases) and depression (22% cases) were reported among the 18 persons who entered their gender as "other". The demographic group with the highest rate of anxiety were those under the age of 35 (52%, 128 cases). The demographic group with the highest rate of depression were those with pre-pandemic personal income below $40,000 per year (18%, 53 cases).

Poor self-reported health was related to higher anxiety and depression, e.g., those with poor self-rated health had 55% rate of anxiety (82 cases) and 28% rate of depression (42 cases) compared to those in good to excellent health, whose rates of anxiety and depression were 36% and 10%, respectively. With respect to health during epidemic, (a) having felt unwell for two or more consecutive days and (b) the belief that the person was infected with virus that causes COVID-19, were associated with increased anxiety and depression. Among 14 persons who reported having been diagnosed with COVID-19, some of the highest levels of anxiety were

**Table 1. Demographics and self-reported health in relation to rates (in %) of Hospital Anxiety and Depression Scale scores above ≥11 (case) plus mean and standard deviations (SD) of the scores among 1,293 participants in survey of adult residents of Philadelphia, PA, during the first wave of COVID-19 epidemic.**

| | | Total | Anxiety | | | | Depression | | | |
|---|---|---|---|---|---|---|---|---|---|---|
| | | | case | | mean | SD | case | | mean | SD |
| | | N | N | % | | | N | % | | |
| **Demographics** | | | | | | | | | | |
| **Gender** | Female | 949 | 388 | 41 | 9.7 | 4.2 | 121 | 13 | 6.4 | 3.8 |
| | Male | 326 | 97 | 28 | 8.2 | 4.3 | 34 | 10 | 5.6 | 3.8 |
| | Other | 18 | 9 | 50 | 11.7 | 4.8 | 4 | 22 | 8.1 | 3.8 |
| **Race** | White | 1105 | 439 | 40 | 9.5 | 4.2 | 137 | 12 | 6.3 | 3.8 |
| | Black or African American | 94 | 26 | 28 | 8.2 | 4.9 | 7 | 7 | 5.5 | 3.6 |
| | Other | 94 | 29 | 31 | 8.9 | 4.3 | 15 | 16 | 6.5 | 4.0 |
| **Age (years)** | <35 | 246 | 128 | 52 | 10.7 | 4.2 | 34 | 14 | 6.7 | 3.7 |
| | 35–54 | 493 | 211 | 43 | 9.9 | 4.2 | 65 | 13 | 6.5 | 3.7 |
| | 55+ | 554 | 155 | 28 | 8.3 | 4.1 | 60 | 11 | 5.8 | 3.8 |
| **Personal income in 2019** | <40,000 | 298 | 138 | 46 | 10.3 | 4.5 | 53 | 18 | 7.1 | 3.9 |
| | 40,000-<100,000 | 596 | 236 | 40 | 9.4 | 4.2 | 72 | 12 | 6.1 | 3.7 |
| | 100,000+ | 365 | 107 | 29 | 8.6 | 4.1 | 28 | 8 | 5.7 | 3.6 |
| | missing | 34 | 13 | 38 | 8.4 | 4.8 | 6 | 18 | 6.0 | 4.0 |
| **Education** | College degree | 1088 | 414 | 38 | 9.4 | 4.2 | 126 | 12 | 6.2 | 3.7 |
| | No college degree | 198 | 79 | 40 | 9.4 | 4.9 | 33 | 17 | 6.5 | 4.2 |
| | Missing | 7 | 1 | 14 | 6.6 | 6.4 | 0 | 0 | 3.3 | 3.3 |
| **Marital status** | Married, or living as married | 755 | 292 | 39 | 9.4 | 4.1 | 81 | 12 | 6.2 | 3.7 |
| | Single | 366 | 152 | 42 | 9.6 | 4.6 | 49 | 13 | 6.4 | 4 |
| | Widowed, divorced | 164 | 46 | 28 | 8.5 | 4.1 | 18 | 11 | 6 | 3.7 |
| | Missing | 8 | 4 | 50 | 11.5 | 4.9 | 1 | 13 | 7.3 | 3.5 |
| **Children <18 years living in your household** | Yes | 286 | 106 | 37 | 9.5 | 4.2 | 39 | 14 | 6.3 | 3.6 |
| | No | 1000 | 387 | 39 | 9.3 | 4.3 | 120 | 12 | 6.2 | 3.8 |
| | Missing | 7 | 1 | 14 | 7.1 | 3.6 | 0 | 0 | 4 | 2.4 |
| **Health** | | | | | | | | | | |
| **Were unwell for two or more consecutive days?** | Yes | 307 | 141 | 46 | 10.3 | 3.8 | 43 | 14 | 6.9 | 3.5 |
| | No | 982 | 352 | 36 | 9.1 | 4.4 | 115 | 12 | 6.0 | 3.8 |
| | Missing | 4 | 1 | 25 | 7.3 | 4.6 | 1 | 25 | 6.3 | 4.2 |
| **Believe infected** | Yes | 98 | 51 | 52 | 10.6 | 4.6 | 17 | 17 | 6.9 | 3.7 |
| | No | 652 | 271 | 39 | 9.5 | 4.2 | 85 | 12 | 6.2 | 3.8 |
| | Maybe | 178 | 85 | 48 | 10.2 | 3.8 | 24 | 13 | 6.7 | 3.6 |
| | Missing | 325 | 87 | 27 | 8.3 | 4.3 | 33 | 10 | 5.8 | 3.9 |
| **COVID-19 diagnosis** | Reported | 14 | 8 | 57 | 11 | 4.4 | 3 | 21 | 6.8 | 4 |
| | Not reported | 1279 | 486 | 38 | 9.3 | 4.3 | 156 | 12 | 6.2 | 3.8 |
| **Self-rated health compared to others** | Good to excellent | 1144 | 412 | 36 | 9.1 | 4.2 | 117 | 10 | 6 | 3.7 |
| | Poor or fair | 149 | 82 | 55 | 11.4 | 4.2 | 42 | 28 | 8.2 | 4 |
| **Self-rated health** | Good to excellent | 1213 | 452 | 37 | 9.3 | 4.3 | 137 | 11 | 6.1 | 3.7 |
| | Poor or fair | 80 | 42 | 53 | 11.1 | 4.5 | 22 | 28 | 7.8 | 4.2 |
| **Upset by experience people trying to avoid you in the public places** | | | | | | | | | | |
| | Yes | 117 | 59 | 50 | 11.2 | 4.3 | 20 | 17 | 7.6 | 3.8 |
| | No | 900 | 331 | 37 | 9.2 | 4.1 | 97 | 11 | 6 | 3.6 |
| | Do not want to answer | 8 | 5 | 63 | 11.0 | 5.9 | 1 | 13 | 7.4 | 3.5 |
| | Maybe avoided by other | 196 | 68 | 35 | 9.3 | 4.3 | 22 | 11 | 6.3 | 3.7 |
| | Did not experience | 261 | 98 | 38 | 9.1 | 4.6 | 41 | 16 | 6.4 | 4.2 |
| **Phase of restrictions during interview** | Most restrictive (early) | 712 | 259 | 36 | 9.1 | 4.3 | 78 | 11 | 5.9 | 3.8 |
| | Restrictions relaxed (late) | 581 | 235 | 40 | 9.7 | 4.3 | 81 | 14 | 6.6 | 3.8 |

seen (57%, 8 cases), but excess of depression was less pronounced (21%, 3 cases), being lower than that mong persons in poor self-reported health (28%). All 14 persons who reported diagnosis of COVID-19 were employed and we reported on them in detail in our earlier analysis that explored occupational factors [9]. Persons who reported that they noticed people avoiding them in public and were upset by the experience were more likely to be anxious (50 vs 38%) but not depressed (17 vs 16%) compared to people who did not report such experiences. We note that the highest rate of anxiety (63%) was among 8 persons who did not want to disclose whether they were upset by the experience, suggesting reluctance to report a traumatic event.

Rates of both anxiety and depression were higher during the later phase of the epidemic, when restrictions were relaxed. Time trends in HADS scores are depicted in Fig 1, with splines fitted to 25[th], 50[th] and 75[th] percentiles of daily distribution of scores, for anxiety (Fig 1a, top) and depression (Fig 1b, bottom), respectively. It is apparent that higher scores were more likely to occur later in data collection as seen in the 75[th] percentile attaining ever higher value; the trend in median is less pronounced; there is a suggestion of widening of the inter-quartile range in time. There was no evidence of abrupt change in pattern after relaxation of stay-at-home order on day 87, the vertical line. The greater excess of cases of anxiety compared to

**(a)**

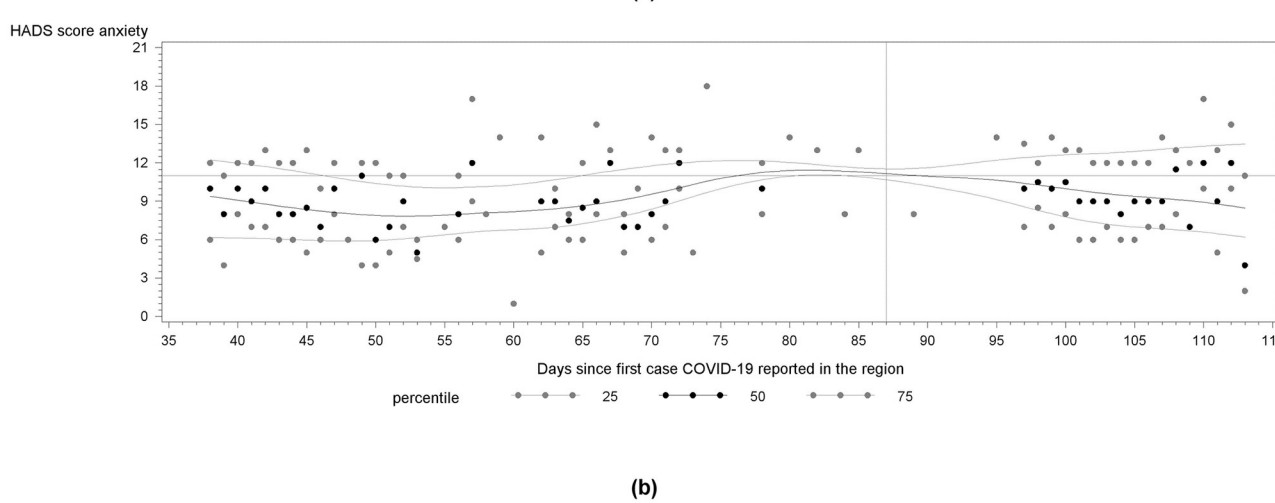

**(b)**

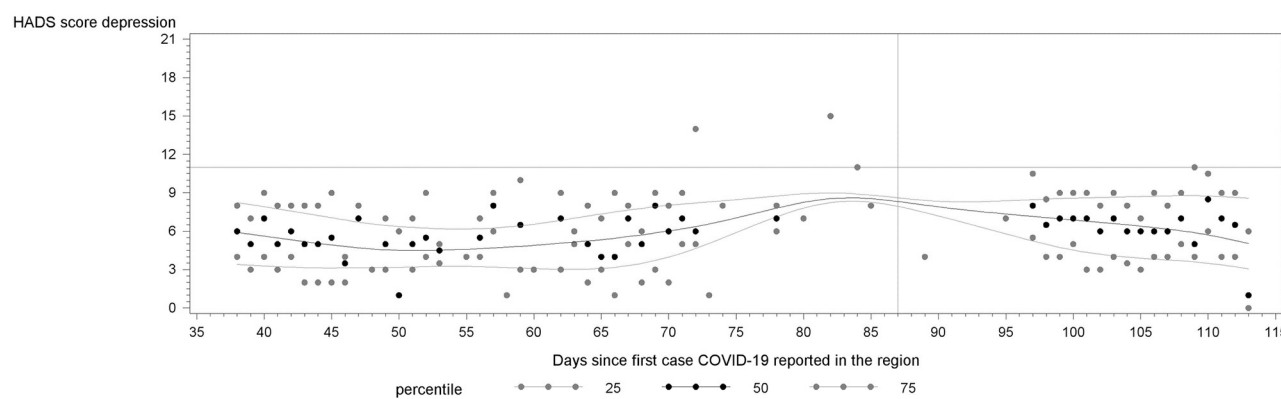

**Fig 1. Time trends in daily medians and inter-quartile range of Hospital Anxiety and Depression Scale (HADS) scores for anxiety (a) and depression (b) since March 10, 2020, the date of data collection since diagnosis of the first COVID-19 case in Philadelphia, PA; horizonal line denotes score of 11 that is used to define a case; vertical line denotes day 87, when restriction on lock-down were relaxed.**

depression is displayed by more of the upper percentiles of anxiety exceeding value of 11, especially in the later dates of data collection.

Precautions taken by study participants were summarized in S2 Table. *Since the start of the epidemic*, most of the participants (84%) used some sort of personal protective equipment (PPE) at home or outdoors, with a "face mask" being the dominant type and gowns being rare. Individuals who resorted to eye protection and gloves were somewhat more anxious and depressed; curiously, persons who did not report PPE use were somewhat more depressed but had similar anxiety scores to the rest. Most participants reported to have taken the following precautions to reduce their risk of exposure to COVID-19: stocked up on essentials at a grocery store or pharmacy (84%), avoided leaving house for non-essential reasons (83%), used social distancing when out in public (94%), avoided crowds and large gatherings (72%). The less common reported precautions included having planned for communicating with family, friends and neighbours (43%), made a plan for caring for others who are ill (18%; most likely indicated of having a person to care for), washing hands more regularly (56%), avoiding touching face (37%), cancelling travel (26%), working from home (27%). Only two persons reported to have taken no precautions. Within this cluster of disclosed precautions, the report of having made plans to care for others who are ill was related to the highest rates of anxiety (46%) and depression (16%). Overall, more anxious individuals tended to report having taken more precautions; the pattern was less distinct for depression. With respect to *behaviours within a week of the response to our questionnaire*, most "went shopping at the grocery store or drugstore" 1 to 3 times or never, and less than half resorted to deliveries of food and medications, the vast majority did not travel to work and avoided social functions, most went outside 4 times or more. Anxiety and depression were related not going to shops, relying on delivery services for food and medications, increased frequency of travel to work, and less frequent trips outside one's dwelling. We stress that these behaviours can be both causes and consequences of mood disorders (a matter that our cross-sectional study design cannot address) and therefore are presented merely to give a fuller description of our participants, not to draw inferences.

Table 2 presents correlations of perceived specific supports and worries, as well as coded most difficult experiences with respect to anxiety and depression scores. Family was the dominant source of perceived support, with a mean score of 80±27, with the next closest being personal doctor with a mean score of 45±29. Federal government was perceived as the least common sources of support with a mean score of 23±20. The higher degree of faith in most reported sources of support were related to reduced anxiety and depression, with family, physician, religious organizations, and neighbours suggesting the strongest effects. There was no such evidence of protective effect among those who expected more support from the city of Philadelphia and social services organizations. The higher level of worries was related to greater anxiety and depression, issue that is explore more in-depth in factor analysis that follows. The dominant worries were about infection of self and family, with means scores of 62 ±27 and 61±32, respectively. Being short of food and medicine were the least prominent worries, with means scores of 31±25 and 31±26, respectively. Among person reporting specific difficult experiences, those that were related to change in work by 248 (19.2%) were associated with increased scores for anxiety (biserial correlation $r = 0.04$, $p = 0.1$) but not depression. The 93 people (7.2%) whose most difficult experiences reflect loss of income were more likely to show symptoms of both anxiety ($r = 0.05$, $p = 0.06$) and especially depression ($r = 0.10$, $p = 0.001$). Experiences that were linked to fear of infection and unavoidable proximity to strangers of 129 persons (10%) were related to heightened anxiety ($r = 0.07$, $p = 0.01$) but not depression. Difficult experiences related to poor health (self or others) was reported by 139 persons (10.8%) and was associated with both mostly anxiety ($r = 0.11$, $p<0.0001$) but also depression ($r = 0.07$, $p = 0.01$). Those troubled by uncertainty (107, 8.3%) were more likely to

**Table 2. Perceived sources of support, worries about the epidemic, and themes of reported most difficult experiences in relation to Hospital Anxiety and Depression Scale scores among 1,293 participants in survey of adult residents of Philadelphia, PA, during the first wave of COVID-19 epidemic; means and standard deviations (SD) of support and worry scores are shown, as well as frequency (in %) of reported themes of most difficult experiences.**

| | Mean | SD | rank correlation (p-value) | | | |
| --- | --- | --- | --- | --- | --- | --- |
| | | | anxiety | | depression | |
| **Where you will find support (no support at all = 0, very strong support = 100)** | | | | | | |
| My immediate family | 80 | 27 | -0.15 | < .0001 | -0.19 | < .0001 |
| My doctor | 45 | 29 | -0.09 | 0.001 | -0.12 | < .0001 |
| Federal Government | 23 | 20 | -0.07 | 0.01 | -0.06 | 0.03 |
| City of Philadelphia | 36 | 24 | 0.01 | 0.81 | -0.03 | 0.27 |
| Department of Public Health (City) | 39 | 25 | -0.02 | 0.50 | -0.08 | 0.004 |
| My religious community | 38 | 28 | -0.07 | 0.01 | -0.11 | < .0001 |
| Social services organization | 25 | 21 | 0.003 | 0.91 | -0.01 | 0.77 |
| My neighbors | 42 | 27 | -0.07 | 0.01 | -0.12 | < .0001 |
| Other | 63 | 20 | -0.02 | 0.43 | -0.06 | 0.05 |
| **Worries about the COVID-19 epidemic (not at all worried = 0, very worried = 100)** | | | | | | |
| I will be infected | 62 | 27 | 0.23 | < .0001 | 0.19 | < .0001 |
| I will infect my family | 61 | 32 | 0.24 | < .0001 | 0.15 | < .0001 |
| I will not be able to cope with the work | 38 | 28 | 0.30 | < .0001 | 0.22 | < .0001 |
| I will become poor | 40 | 30 | 0.30 | < .0001 | 0.23 | < .0001 |
| I will be short of food | 31 | 25 | 0.26 | < .0001 | 0.18 | < .0001 |
| I will be short of medicines | 31 | 26 | 0.22 | < .0001 | 0.19 | < .0001 |
| I will fail myself and my family | 41 | 29 | 0.35 | < .0001 | 0.24 | < .0001 |
| I will be confined at home and not able to leave | 48 | 30 | 0.25 | < .0001 | 0.23 | < .0001 |
| **Themes of most difficult experiences (present or absent)** | N | % | biserial correlation (p-value) | | | |
| Change in work: self or family | 248 | 19.2 | 0.04 | 0.1 | -0.003 | 0.9 |
| Change in schooling | 7 | 0.5 | 0.02 | 0.5 | 0.03 | 0.2 |
| Childcare | 88 | 6.8 | -0.01 | 0.8 | 0.02 | 0.5 |
| Lost income (self or family) | 93 | 7.2 | 0.05 | 0.06 | 0.10 | 0.001 |
| Unavoidable proximity to strangers/ fear of infection | 129 | 10.0 | 0.07 | 0.01 | 0.04 | 0.2 |
| Social isolation; cabin fever | 126 | 9.7 | -0.04 | 0.2 | 0.01 | 0.7 |
| Poor health self or others | 139 | 10.8 | 0.11 | < .0001 | 0.07 | 0.01 |
| Trouble sleeping | 9 | 0.7 | 0.05 | 0.08 | 0.01 | 0.6 |
| Uncertainty | 107 | 8.3 | 0.05 | 0.05 | 0.04 | 0.1 |
| Media coverage | 15 | 1.2 | 0.001 | 1.0 | -0.004 | 0.9 |
| Food access | 9 | 0.7 | -0.003 | 0.9 | -0.002 | 0.9 |
| Death family or friend | 32 | 2.5 | 0.05 | 0.05 | 0.04 | 0.1 |
| Anger | 10 | 0.8 | -0.01 | 0.8 | 0.02 | 0.6 |

show symptoms of both anxiety (r = 0.05, p = 0.05) and depression (r = 0.04, p = 0.1). Death of family member of a friend (32, 2.5%) was associated with elevated signs of both anxiety (r = 0.05, p = 0.05) and depression (r = 0.04, p = 0.1). Most difficult reported experiences that were related to changes in schooling, childcare, social isolation, trouble sleeping, media coverage, food access, and anger did not correlate with HADS scores; we prefer not to over-interpret this lack of statistical associations as these experiences not contributing to mental health. The coded themes showed little inter-relation among each other in exploratory likelihood factor analysis with the largest eigenvalue barely above 1 and therefore will be treated as independent in the subsequent regression analysis. The patterns are not materially different for the minority

(n = 369) who did not have a job at the start of the pandemic and their statistics are provided in S3 Table.

## Factor analysis of worries and HADS scores

Factor analysis that examined interrelation of specific worries and HADS scores revealed that three correlated factors were sufficient to explain the latent structure ($\chi^2_{(18)}$ = 136, p <0.0001). The main observations are summarized in the path diagram, Fig 2, associated scoring coefficients that can be used to predict measures of latent factors from the observed data are given in Table 3. Detailed output of factor analysis including scree plot and visualization of the rotated factor patterns are exhibited in **Tables and Figures** in S4 Table. As expected, given established validity of HADS scores for anxiety and depression, they were associated with factor 2, which we termed "mood disorders". These were positively related to two groups of specific worries. The first one was about hardships related primarily to necessities of life, such as food (loading 0.79), medicine (0.72), and money (0.50). The lesser component of worries about hardships was related to trepidations regarding one's ability to meet one's own expectation of conduct during the epidemic: coping with work (0.31) and "failing oneself and family" (0.43). The other latent worry was that of fear of infection of oneself and family, with fear of infecting others (family) dominating, with loading of 0.94. There was a positive association between worry about hardships and fears of infection. Worry about being confined to home was not related to any of the latent factors as evidenced by a factor loading <0.3; the p-values for these loading not even approaching the conventional cut-off of <0.05. We created factors scores for "hardships" and "fear of infection" to further explore their correlates in regression analyses.

## Regression analyses

All multiple regression analyses were adjusted for gender, race, age, personal income, education, marital status, children <18 at home; we also further adjusted for employment status and self-reported health. We do not display estimates of regression coefficients for these adjustment factor because they largely agreed with descriptive analyses, and we wish to focus instead on specific experiences during the epidemic. Adjustment for these variables had no material impact on the associations depicted in Table 4.

We first describe results related to coded themes of the most difficult experiences that the participants shared with us. Loss of income was associated with elevated anxiety, depression, and worry about hardships, but not fear of infection. The reports of unavoidable proximity to other people were related to both anxiety and fear of infection. Those who reported difficulties with social isolation appeared to be less fearful of infection. Experiences with poor health were related to both anxiety and depression, as well as fear of infection, but not worry about hardships. Uncertainty about what is to come and death of a family member or a friend were independently related to both anxiety and depression, but not the specific worries. Those who reported to have struggled with accessing food showed higher worries about hardship only. The experiences that mapped into the following themes were not related to the outcomes considered in regression models: change in work: self or family, change in schooling, difficulties with childcare, trouble sleeping, media coverage, and anger.

There were no notable interactions of gender (male vs. female) by coded themes of most difficult experiences on level of anxiety (see **Tables in** S5 Table). Change in schooling was related to greater increase in depression and worries about hardships among men, but only two were affected, leaving chance the most plausible explanation, despite p = 0.01 and p = 0.03 for interaction, respectively. There is a suggestion of main effect of change in schooling on increased depression but not hardships (Table 4). Excess of depression among those who

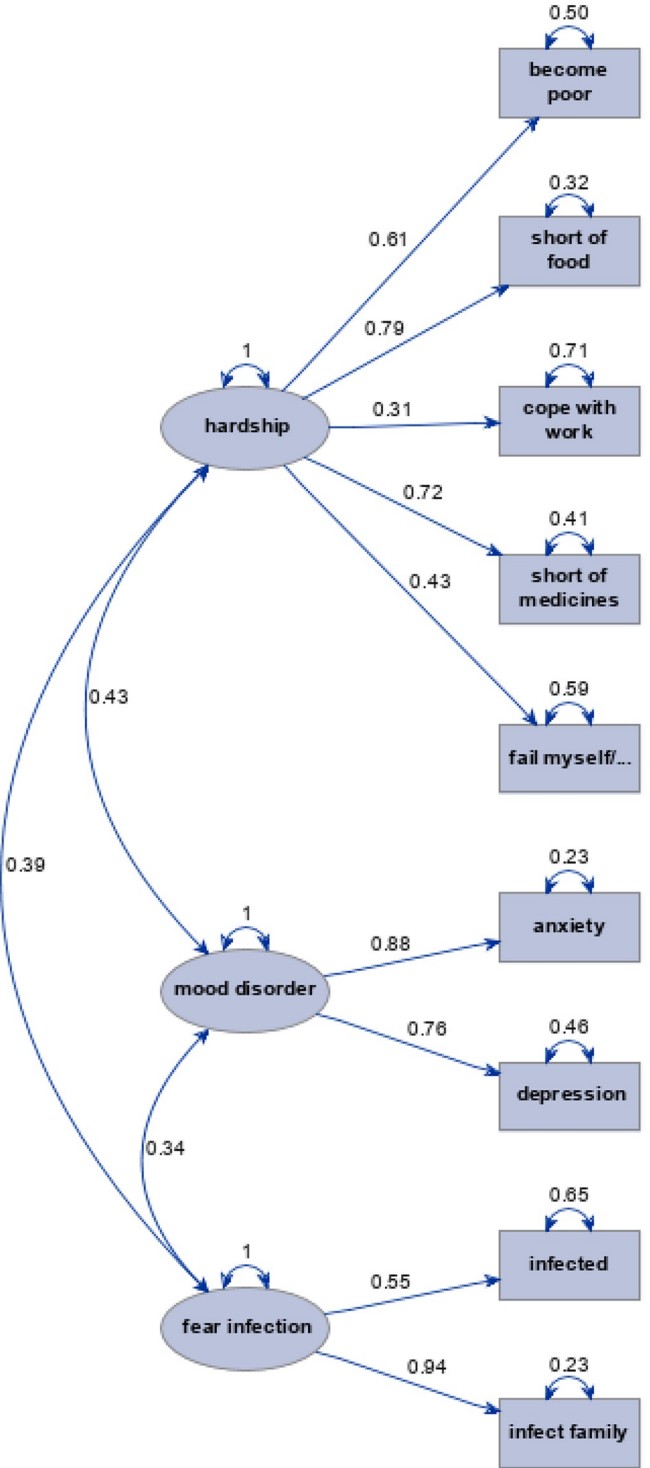

**Fig 2. Directed paths with loadings greater than 0.3, p<0.001: Inter-relation of HADS scores and specific worries.**

**Table 3. Standardized scoring coefficients for the path diagram (Fig 2) of factor analysis of worries and Hospital Anxiety and Depression Scale (HADS) scores: Maximum value per factor is given in bold.**

| Variable recorded from the 1,293 participants | Label in Path Diagram | Factor 1: "Worry about hardship" | Factor 2: "Mood disorder" | Factor 3: "Fear of infection" |
|---|---|---|---|---|
| HADS anxiety score | anxiety | -0.02095 | **+0.622** | +0.0532 |
| HADS depression score | depression | -0.00842 | **+0.27059** | -0.00664 |
| I will be infected | infected | 0.0302 | +0.0085 | **+0.14958** |
| I will infect my family | infect family | -0.0778 | -0.0206 | **+0.7119** |
| I will not be able to cope with the work | cope with work | **+0.07759** | +0.04627 | +0.05606 |
| I will become poor | become poor | **+0.20913** | +0.04406 | +0.04013 |
| I will be short of food | short of food | **+0.41503** | +0.0361 | +0.05076 |
| I will be short of medicines | short of medicines | **+0.3007** | +0.00846 | +0.07235 |
| I will fail myself and my family | fail myself/family | **+0.1293** | +0.06659 | +0.05342 |
| I will be confined at home and not able to leave | confined at home | **+0.05334** | +0.04094 | +0.01585 |

reported death of a either a family member or a friend was driven by seven men with marked increase in mean HADS scores for depression (9.3 in affected vs 5.5 in unaffected), p = 0.04 for interaction. Women who reported difficulties related to lost income (N = 70) were more fearful of infection (factor score in Fig 2), but men (N = 23) were less fearful (none of the persons with gender "other" reported such experiences); on average, women were more fearful; p = 0.049 for interaction.

Effect of themes of specific "difficult experiences" were independent of the estimated effect of having been upset by people trying to avoid one in public, which was assessed directly and systematically. Those who were upset by such experiences were more likely to be anxious and depressed. Whether they reported to having been upset by the experience or not, people who noticed others avoiding them in public were more likely to score higher on fear of infection, but not hardship.

Regression analyses confirmed that the risk of anxiety and depression increased independently of all other factors considered with the duration of the epidemic during the studied period, as depicted in Fig 1, with p-values for the effect estimates <0.0001. The estimated associations with worries about hardships did not alter in time, but fear of infection increased, with a convincing p<0.0001.

Among supports that alleviated risks, immediate family featured most prominently, conferring protection against anxiety, depression, and hardships but not fear of infection. Persons who reported expecting seeking support from social services organizations were worse off on all outcomes, possibly reflecting their pre-epidemic vulnerability rather than reflecting on effectiveness of such organizations in providing relief. Worries about hardships and infection were higher among those who could rely on their doctors, possibly because they were also in poor health and our self-reported health questions did not capture the full complexity of such needs. The reported reliance on support from the Federal government was linked with worry about hardships. Persons reporting that they seek support from city of Philadelphia were more depressed and worried about hardships and infection. However, report of relying on the city's Department of Public Health appeared to be related to lower depression. The reported support from religious community likewise was linked to reduced depression but elevated fear of infection. Having supportive neighbours was related to lower symptoms of depression.

**Table 4.** Estimated coefficients of multiple regression models relating experiences of 1,293 participants in survey of adult residents of Philadelphia, PA, during the first wave of COVID-19 epidemic to (a) Hospital Anxiety and Depression Scale scores via negative binomial regression and (b) measures of hardships and fear of infection assessed by factor scores computed from reported specific worries via linear regression; RR: relative rates of change in one unit on the scale; CI: confidence intervals; p: p-values for the test of no association; all displayed estimates are mutually adjusted and account for gender, race, age, personal income, education, marital status, children <18 at home (there was no material change on further adjustment for employment, self-reported general health; comments on effect modification by gender are in text).

| Predictor variables | | Anxiety | | | | Depression | | | | Hardship (factor score) | | | | Fear of infection (factor score) | | | |
|---|---|---|---|---|---|---|---|---|---|---|---|---|---|---|---|---|---|
| | | RR | 95% CI | | p | RR | 95% CI | | p | estimate | 95% CI | | p | estimate | 95% CI | | p |
| Themes of most difficult experiences | Change in work: self or family | 1.02 | 0.96 | 1.08 | 0.55 | 0.99 | 0.91 | 1.08 | 0.88 | -0.05 | -0.17 | 0.08 | 0.46 | 0.05 | -0.07 | 0.17 | 0.45 |
| | Change in schooling | 1.14 | 0.83 | 1.57 | 0.41 | 1.36 | 0.89 | 2.07 | 0.15 | -0.04 | -0.67 | 0.59 | 0.91 | 0.15 | -0.49 | 0.78 | 0.65 |
| | Childcare | 0.95 | 0.85 | 1.05 | 0.31 | 1.07 | 0.93 | 1.24 | 0.36 | -0.15 | -0.36 | 0.06 | 0.16 | -0.12 | -0.33 | 0.09 | 0.27 |
| | Lost income (self or family) | 1.09 | 0.99 | 1.19 | 0.08 | 1.23 | 1.09 | 1.39 | 0.001 | 0.41 | 0.23 | 0.59 | <.0001 | 0.02 | -0.16 | 0.20 | 0.82 |
| | Unavoidable proximity to strangers/ fear of infection | 1.09 | 1.01 | 1.18 | 0.02 | 1.07 | 0.96 | 1.19 | 0.21 | 0.13 | -0.03 | 0.28 | 0.10 | 0.29 | 0.13 | 0.44 | 0.0003 |
| | Social isolation; cabin fever | 0.95 | 0.88 | 1.04 | 0.26 | 1.05 | 0.94 | 1.18 | 0.35 | -0.10 | -0.26 | 0.06 | 0.21 | -0.22 | -0.38 | -0.07 | 0.01 |
| | Poor health self or others | 1.14 | 1.06 | 1.23 | 0.001 | 1.14 | 1.03 | 1.27 | 0.01 | 0.04 | -0.11 | 0.19 | 0.59 | 0.18 | 0.03 | 0.33 | 0.02 |
| | Trouble sleeping | 1.19 | 0.91 | 1.56 | 0.21 | 1.09 | 0.74 | 1.59 | 0.67 | -0.03 | -0.58 | 0.53 | 0.93 | -0.07 | -0.63 | 0.49 | 0.80 |
| | Uncertainty | 1.08 | 0.99 | 1.17 | 0.10 | 1.12 | 0.99 | 1.25 | 0.07 | 0.06 | -0.11 | 0.23 | 0.48 | -0.01 | -0.18 | 0.16 | 0.92 |
| | Media coverage | 1.02 | 0.82 | 1.28 | 0.84 | 0.96 | 0.71 | 1.30 | 0.81 | 0.10 | -0.33 | 0.53 | 0.63 | 0.16 | -0.28 | 0.59 | 0.48 |
| | Food access | 1.01 | 0.75 | 1.35 | 0.96 | 1.00 | 0.68 | 1.49 | 0.99 | 0.51 | -0.04 | 1.07 | 0.07 | 0.26 | -0.30 | 0.82 | 0.37 |
| | Death family or friend | 1.13 | 0.97 | 1.31 | 0.11 | 1.18 | 0.96 | 1.45 | 0.11 | 0.01 | -0.29 | 0.31 | 0.92 | -0.04 | -0.34 | 0.26 | 0.78 |
| | Anger | 1.03 | 0.78 | 1.37 | 0.82 | 1.15 | 0.80 | 1.65 | 0.45 | -0.09 | -0.62 | 0.44 | 0.74 | 0.08 | -0.45 | 0.61 | 0.77 |
| Upset by avoided in the public places | Do not want to answer | 1.06 | 0.78 | 1.44 | 0.69 | 1.11 | 0.74 | 1.68 | 0.62 | 0.13 | -0.48 | 0.73 | 0.69 | 0.18 | -0.43 | 0.79 | 0.56 |
| | No | 1.00 | 0.94 | 1.06 | 0.94 | 0.94 | 0.86 | 1.02 | 0.12 | -0.02 | -0.13 | 0.10 | 0.80 | 0.17 | 0.05 | 0.28 | 0.01 |
| | Yes | 1.20 | 1.09 | 1.32 | 0.0002 | 1.15 | 1.01 | 1.30 | 0.03 | 0.07 | -0.12 | 0.25 | 0.48 | 0.22 | 0.04 | 0.41 | 0.02 |
| | Did not experience | ref. | | | | ref. | | | | ref. | | | | ref. | | | |
| Days since start of epidemic (per 30 days) | | 1.06 | 1.03 | 1.09 | <.0001 | 1.09 | 1.05 | 1.13 | <.0001 | 0.03 | -0.02 | 0.08 | 0.23 | 0.16 | 0.11 | 0.21 | <.0001 |
| Supports (per 25 units) | My immediate family | 0.95 | 0.93 | 0.97 | <.0001 | 0.92 | 0.89 | 0.95 | <.0001 | -0.11 | -0.15 | -0.06 | <.0001 | 0.02 | -0.03 | 0.07 | 0.42 |
| | My doctor | 0.99 | 0.97 | 1.02 | 0.60 | 0.98 | 0.94 | 1.01 | 0.15 | 0.04 | -0.01 | 0.09 | 0.10 | 0.05 | 0.00 | 0.09 | 0.07 |
| | Federal Government | 0.98 | 0.95 | 1.01 | 0.18 | 1.00 | 0.96 | 1.04 | 0.97 | 0.07 | 0.01 | 0.14 | 0.02 | 0.0 | -0.06 | 0.06 | 0.99 |
| | City of Philadelphia | 1.03 | 0.99 | 1.06 | 0.17 | 1.06 | 1.01 | 1.11 | 0.02 | 0.07 | 0.00 | 0.14 | 0.04 | 0.08 | 0.01 | 0.15 | 0.02 |
| | Department of Public Health (City) | 1.00 | 0.97 | 1.04 | 0.93 | 0.95 | 0.91 | 1.00 | 0.05 | -0.01 | -0.07 | 0.06 | 0.82 | 0.04 | -0.02 | 0.11 | 0.22 |
| | My religious community | 0.98 | 0.96 | 1.01 | 0.17 | 0.95 | 0.92 | 0.99 | 0.005 | 0.02 | -0.03 | 0.07 | 0.41 | 0.05 | 0.00 | 0.09 | 0.06 |
| | Social services organization | 1.03 | 1.00 | 1.06 | 0.08 | 1.05 | 1.00 | 1.10 | 0.05 | 0.15 | 0.09 | 0.22 | <.0001 | 0.07 | 0.01 | 0.14 | 0.04 |
| | My neighbors | 0.99 | 0.96 | 1.01 | 0.28 | 0.97 | 0.93 | 1.00 | 0.05 | -0.03 | -0.07 | 0.02 | 0.31 | -0.01 | -0.06 | 0.04 | 0.79 |
| | Other | 1.00 | 0.97 | 1.03 | 0.84 | 0.97 | 0.93 | 1.01 | 0.11 | 0.08 | 0.02 | 0.14 | 0.01 | 0.01 | -0.05 | 0.07 | 0.70 |

## Thematic analysis

In addition to the themes coded as present or absent and used in quantitative analysis above, the investigators believe that the following cross-cutting themes emerged from examination of free text narratives supplied by the 782 respondents (about a half; 775 had complete data and were also included in regression analyses above). Description of the 775 participants who provided narrative responses and were included in the regression analyses is given in S1 Table. There were no stark differences among sub-samples, except that there may be some excess of anxiety but not depression among those who contributed free text narratives.

**Changes to work: Hardship for most but better for some.**   Some respondents found that their working arrangements improved as the result of stay-at-home orders, e.g., (a) by better accommodating their disability, and they dread losing these advantages, (b) "this epidemic is the best thing that has happened to me. I am making money hand over fist working extra shifts at the hospital." However, most respondents who were forced to work from home reported that "working from home in general is very stressful and mentally exhausting".

Loss of childcare was the most difficult experience during the epidemic for working and not working people alike:

> "I DON'T HAVE A WORK. I DON'T HAVE MONEY. I CAN'T SEND MY KID TO A DAYCARE BECAUSE EVERYTHING IS CLOSED";

> "feeling like I am living at work and not working from home".

Dealing with customers at work has become more challenging for some, as a by-product of stay-at-home order:

> "all the normal people are quarantined we have had to deal with really bad people at the front desk, like drug users, crooks, mentally insane, people that just want to argue and start fights, etc."

The need to take public transit to commute to work has become a challenging daily experience for some:

> "Homeless and addicts on SEPTA. Rude people on SEPTA smoking and showing nor regard for others health or safety".

Our survey did not capture experiences of people with multiple jobs, only some of which were not eliminated by stay-at-home orders, as pointed out by one respondent who conveyed their sense of loss of autonomy and purpose:

> "in my case the [job] I love and have done my whole adult life [. . .], has been taken from me due to covid. It defines me and is my calling. My day job actually has increased to full hours to provide me with a living wage. I should be grateful but unfortunately I feel more depressed. It is like going through war, with an unknown enemy we don't know when this enemy will be defeated, . . .".

Some reported that it was difficult to not have their work-related hardships appreciated by others:

> "Wearing face masks for my entire shift and listening to the constant complaining about having to wash their hands and wear masks."

Media coverage of pandemic was a prominent stressor for some respondents, whether they were its consumers or producers.

Several individuals shared that one of the biggest sources of stress was due to perceived poor leadership from their employers: e.g., "my boss is being more of a jerk than usual", "my boss [is] unsupportive and has the emotional range of a lizard". However, the stress of leadership is not be under-estimated, as some of the most difficult experiences related to effects on others, not self: "I have to furlough everyone who works for me." The burden of leadership and responsibility for one's employees weighed most heavily on some who had to ensure "staff remain protected at work."

**Stay-at-home orders: The economic toll (it affects health too).** Loss of health insurance, related to loss of employment and/or lack of suitable government programs, weighed heavily on some:

> "No insurance and then not reopening the marketplace and being denied Medicaid has left me terrified of getting sick after seeing bills for $1 million online."

> "The entire industry has shut down. Nobody has any idea when it will restart. People will have problems qualifying for health care."

Not being able to get essential medical care due to shutdown is an understandable source of reported stress aggravated by financial hardships:

> "I am experiencing a . . . recurrence of [cancer] . . . stress not knowing when I'll feel safe treating this. Also stressed my dentist office closed for good, due to financial losses from Covid 19 closure, and they owe me a great deal of money."

Economic impact of management of the epidemic threatened some with homelessness:

> "One of my roommates has lost her job due to shutdowns and may not be able to pay rent in a couple of weeks, and that may or may not affect my own housing situation as well."

> "I'm frustrated about not being able to apply for unemployment and afraid I'll end up homeless if I don't start making my normal amount of income."

> "I'm worried that if I get laid off, we will become homeless."

Respondents voiced dissatisfaction with the breadth of stay-at-home orders that were perceived as unjustified:

> "To destroy jobs and force people out of work when they are in the prime earning years of their lives is abhorrent especially when they have a near zero risk of dying from this. . . . People over the age of 55 should continue to shelter in place at their own discretion."

Some small business owners suffered and felt abandoned by government agencies:

> "The fact that the Commonwealth of PA shut down the economy with no regard to small business's [sic] is a disgrace. . . . I care about the business I have built. I care about my employees. I care about my customers. Why does the government not care about me? The government has failed small business & the self employed [sic]."

**Stay-at-home orders: Psychological wins and (mostly) heavy blows.** The slower pace of life imposed by the shutdown were reported as enjoyable "more time with my kids and slower pace of life. I almost never have to drive!", even among people who suffered the "loss of the usual childcare arrangements," a persistent source of difficulty for many. Some people reported that they are "*enjoying* [sic] the quarantine" that led to "extended quiet time home alone, working from home" and "wish we could do this (minus the virus part) all the time." However, the demands of parenting were among most common concerns: "having our children out of school and trying to support them mentally, socially and work while trying to balance these same things for ourselves." The challenge of caring for elderly parents in poor health without the usual support from outside agencies proved a difficult experience. Not being with family for funerals and when family members are dying is a source of most painful experiences:

"My [parent] is dying in a nursing home and no one is allowed to visit. Sometimes . . . too weak to speak on the phone."

The overwhelming narrative of life under stay-at-home orders was negative. Even among people who accepted the importance of social distancing, disruption to social cohesion was a commonly voiced hardship, such as:

"not seeing friends and family has been extremely depressing, as well as being forced to cancel multiple spring and summer vacation plans",

"being at home 99% of the time means my support/social network as shrunken immensely. I often feel I don't matter."

Among people who were supportive of collective actions and personal sacrifices to fight the epidemic, there was exhaustion "from being inside all the time, being made to wear a suffocating mask outdoors (you ever have a panic attack while wearing one?), having . . . skin cracked and itching to the point of blood from washing and washing," indicating that restrictive measures have taken tremendous toll: "honest to god I'm better off dead."

Social isolation was conflated for some with loss of opportunities and hope:

"Most of my anxious feeling stem from being very far away from my family. . . . I was planning on moving back . . . in hopes of finding a good job again. Maybe in time, but I doubt I will ever recover from this and will be stuck here forever to die alone. Sorry, brutally honest here. . ."

The most common long and emotionally charged shared stories by the respondents expressed their sense that protection from infection through social isolation was just not worth it because it was "robbing us of our humanity", even if they themselves were among the highest risk groups for death from COVID-19:

". . . I am really too old to be terribly concerned about getting the virus, but the restrictions we are living under are very difficult to endure. Old people suffer from isolation, & this is major isolation. I miss being close to people . . .. . . . I understand the need for these rules, but it's not a lot of fun."

"I'm a professional singer . . . We have no idea when we'll be able to sing again. Music is my source of spirituality and I can't even bring myself to sing. Crying has even become difficult. Zoom is vile. I feel like this is robbing us of our humanity."

"This just feels unsustainable [sic] and I don't know how we ask people to continue to live like this, but I also feel it's unsafe to open things up again. We need a plan for dealing with this besides "stay home and rot." This is . . . so bad for us mentally and physically, the economy is cratering and the plan going forward is. . . what? There isn't one. People are going to start going nuts, others are going to go out and get sick and things are going to be even worse. It's horrifying."

"Everything I loved—all my social interaction, intellectual and spiritual stimulation—has been robbed from me. The images of a future of plexiglas [sic] and masks and distancing "until there's a vaccine!!!" is worse than an Orwellian novel. Everyone I know is stressed, anxious, can't concentrate. I watch my co-workers deteriorate daily as the stay at home becomes endless, they have no escape from children who are bouncing off the walls, the schools are uncertain for the fall and people have no outlet for anything that brings them joy—working out at the gym, a meal out, a movie, a browse through a store, let alone hobbies like playing music, sports or social groups. The shaming, the snitching, the holier than-thou crowd that tell you are selfish for not wearing a mask make me feel suicidal. I'm just despondent over it all."

"I feel alienated from other living people/neighbors [sic] whom I see outside from my window, who are going about their business. That is a new and alarming feeling for me, as I used to be outgoing, very civic-minded, friendly, cheerful [sic] and optimistic. No doubt due to lack of live human-to-human contact, which I've compensated for with over-much media consumption, I feel on the edge of a nervous breakdown."

"Not being touched for 3 months has been impossible to deal with. I was social distancing for a while, but honestly, I recently eased up on my isolation so that I could have sex again."

"I want to hold my kids!!!"

"I desperately miss physical contact with my grandchildren and taking them on outings."

**Fear of infection.** The fear of infection due to unavoidable interactions with strangers and the overwhelming number of prescribed precautions is illustrated by this shared experience:

"Putting on a mask to go out causes me a slightly anxious feeling at first. I believe this is partly because of a reduced airflow, but also because of having to pay extra attention to the ritual of checklisting [sic] all the social distancing gear: mask, eyewear, gloves, hand sanitizer, tissues [sic] or handkerchief, etc. This feeling does not last on a walk or doing yard work, but increases inside a store. The increased feeling is definitely due to having to work with new kinds of restricted and possibly dangerous or unwanted social interactions."

Fear of infection due to unsafe conduct by others in the community was a common theme: "I'm sad, depressed and very angry that, as more resume their lives, even basic requests to ensure the health of older people are dismissed with a flippant 'If you're so worried, stay home.'"
Healthcare workers endured fears of infection to themselves and their families due to unprotected exposure to infected patients, to whom they felt obliged to still provide care:

"The most stressful event has been having been exposed to a patient with Covid and being sent out of work for a week as I didn't have the proper PPE on."

Healthcare workers appeared to be particularly conflicted between sense of duty to their patients and personal risks:

> "Some of us in the healthcare profession had to make the choice to stop working because of being high risk to covid-19 due to pre-existing conditions . . ., even though our jobs were up and running for business, we had to stop working, which left us feeling guilty and use-less, on top of all the other feelings of fear and worry."

Fear of return to work and uncertainty is captured by sharing that a respondent is "more afraid in the yellow phase because there's so much uncertainty, no clear cut guidrlines [sic], and lots of conflicting information and differing opinions', with their most difficult experience stemming from this uncertainty of "weighing health against money."

**Health concerns: Not just about "me".** Some people expressed inability to help with making the situation better for others as their main challenge. Desire to help others was the priority for some, despite being at considerable risk of poor outcome, if infected:

> "I don't want to get sick because I am very old and so is my [spouse] . . .. But I think ALL the time about the people who are suffering without health care or support and about our city,. which is so poor and was poor before this catastrophe. . . . As soon as the stimulus arri-ves, we shall both give it to Philabnndance [sic] or the National Domestic Workers Union."

> "I'm concerned about the people who aren't doing well. That concern affects me more than my personal needs. . . . People don't just care about themselves."

Concerns for others illustrated by claim that the most difficult experience was "knowing people do not have access to food and losing their jobs."

A recurring concern was not just for one's health but that of others, vividly exemplified con-cerns about "being pregnant and worrying how possible infection may impact my child and birth plan," a mix of concerns about self, one's child and overall uncertainty. For some, con-cerns about one's own health, that of their family and sense of responsibility to others collided to place some into very challenging situation:

> "Being both pregnant and a frontline healthcare worker during this time is really challeng-ing. I feel like my responsibilities to my job/ my patients are at odds with my responsibilities to myself/my family/ my baby."

Persons who think that they are at high risk of infection expressed concern about risk to their family and friends, they shared conflicting emotions about wishing to be with family and friends and yet limiting this because of their professional obligations that entail heightened risk:

> "As a healthcare worker, I feel more isolated from family and friends buy it seems like it's a part of the duty/obligation as I'm exposed more frequently and a higher risk of being a car-rier. I fear the risk I pose to others far more than the risk of me getting sick."

Feeling of guilt for incidentally exposed family to risk due to uncontrolled pressures from work and lack of information on the degree of risk early in the epidemic was among one of the most traumatic experiences for some, exemplified by this story:

> ". . . when my partner was still working and I had to work with my child, I took her to my office and felt so guilty that I had to bring her (her school closed but we were not yet and I

had to get my office things to wfh [sic]) and expose her to the virus even though back in March I had no idea how bad the coronavirus was. Still it was still to this the most stressful day of this moment b/c [sic] I felt work pressures and had my young child with no help."

Believing that one nearly died from COVID-19 and lack of certainty about diagnosis and risk to others due to lack of faith in accuracy of testing was understandably among most difficult experiences: "Almost dying from covid, not knowing if I infected my clients and potentially killed everyone in their apartment building . . ..."

Having a family member fall seriously ill with COVID-19 increased anxiety, made the risk palpable and personal, aggravated by inability to be close to them to help:

"I currently have a [sibling] who lives far away and has Covid-19 and is on a ventilator, so that has increased my anxiety. Before that happened I did not have any anxiety. Now the text ping sound gets me anxious for fear of more bad news."

Given the high risk of death during the epidemic, some "think there should be more public discussion about caring for very elderly relatives at home in the event they contract COVID-19. There seems to be a presumption that they will go to the hospital for medical care where their prognosis is not thought to be good and where they are likely to be isolated from family and even medical caregivers for most of the time. It is unclear why families would pursue this path" and a serious consideration of "adequate palliative care at home." The challenges of palliative care were also highlighted from the perspective of healthcare workers who reported that their most difficult experience was "patients dying alone."

Some expressed mixed feelings of grief and gratitude in relation to experience of others in the community:

"I feel incredibly lucky . . . So much of handling the pandemic has felt like a prolonged grieving experience; for those who are sick, for those who pass, for their families and friends, for the demands and sacrifices we are forcing on our essential workers, for the celebrations that are cancelled, the trips to the Shore that will have to wait until next year, for the summer bounty at the farmers markets that will likely be skipped for safety reasons. The list goes on and on. Holding grief and gratitude together is tough stuff to begin with, but even harder under these circumstances."

**Grievances about government response.** Lack of testing was a commonly voiced grievance, coloured by sense that the government simply did not care about some vulnerable segments of the population, such as low-income seniors:

"The lack of testing.. is unacceptable. . . Low Income Seniors [sic] in particular are being left to fend for themselves with little to no support, information, access to testing and follow up care. A Share box here and there doesn't cut it. But I guess they're expendable."

"I feel that covid19 is a serious infection, however I feel like Pennsylvania/Philadelphia governments are not concerned for the total welfare of their citizens."

Several respondents voiced loss of autonomy as one of the most difficult aspects of the epidemic, voicing their frustrations that they "wish that everyone didn't want to turn this health emergency into a Nanny State." Concerns appeared to arise from perceived authoritative measures taken by government, which undermine personal autonomy, exemplified by this story:

"I would say I was used to "Social Distancing" before it was a 'thing', but, now, with it being nearly "Mandatory"- I find myself wanting to go out; seeking human contact. I DISLIKE wearing a mask; I see SO MANY others' not, I want to be DEFIANT in the face of 'Rules'".

The sense of lack of support and confidence in the epidemic being managed appropriately was exemplified by these comments from a person who struggled to pay rent due to loss of employment:

"I feel like no one has a handle on this situation at any level. . . .. I feel extremely let down by my government and my community, where I often see people without masks hanging out in close contact."

Frustration with lack of help from expected sources and loss of confidence in such agencies being able to help were exemplified by these experiences:

"One thing about COVID-19 you get to see what is truly essential as a service, etc., and what is not. It came to the reality that all the government money that many of the City of Philadelphia non-profits beg for . . . that in a real epidemic they are useless to the residents of the City of Philadelphia. Many of their services at this time are useless. Many . . . haven't thought of any way to get into the communities, remake themselves for an emergency such as COVID-19 . . . how sad is that?"

"I work, well at this point worked, in the City of New York. I have had zero confidence in the governor of New York and mayor of New York. They never instituted a proper lockdown and caused a great deal of pain and suffering. I suspect I had the virus back in January as I had all the symptoms then and have some after effects [sic] now. The antibody tests may not be sensitive enough to pick up an exposure from six months ago. Other people I know in the NYC area suspect the same thing. This was at the point when we were all told not to worry by the powers that be at various levels of government. That sapped any faith I had in them."

Inconsistency in application of social distancing rules and or lack of enforcement was among one of most difficult experiences reported by many, e.g., "Philadelphia officials allowing drug addicts and prostitution to continue . . . and gather in large groups."

Some parents of incarcerated persons feared that the epidemic would be spread there rapidly, and that epidemic was preventing early release and rehabilitation efforts, holding the "system" responsible:

"Biggest concern is [my child] in Philadelphia prison system. Worry about quickly Coronavirus could spread there. He is eligible for release to rehab facility if evaluation could be done but all alcohol and drug evaluations on hold since March."

For some, fear of infection was negligible compared to fear of government over-reach in responding to the epidemic, causing both "excruciating" loneliness and "severe anxiety about events that stemmed from this lockdown, like economic concerns and the fact that there is abuse of power by the government", leading to "fear that the government will shut down again and the aftershocks from those draconian policies will put the nation into economic turmoil". Likewise, for some "political environment" was far more upsetting than the epidemic per se: "Trump is a dictator. The racial tensions, riots, looting and curfews. All these have stressed me out WAY more than Covid."

**Anger, riots, and incivility.**   Our survey did not assess anger and yet it seemed to be a dominant emotion for some respondents: ". . . anger. I've been feeling a lot of that," aimed at those who hold views different to those of the respondents, typically entailing political grievances (evenly aimed at all political parties and ideologies). Anger at people who did not follow rules of conduct that respondent preferred was a common thread, likely related to fear of infection as well pollicised nature of these differences:

"What frustrates and upsets me is the looting and rioting on top of the virus, on top of the 31/2 years of trump [sic] and Republicans destroying our country. And the apparent twisted nature of a third of the population who still support the Republicans, refuse to believe the virus is a concern, and continue to endanger our healthcare."

A common theme for most difficult experience was irritation at the perceived inconsiderate and irrational actions of others, for example:

"throwing dirty/used gloves and masks on the ground. Still don't understand the thought process of someone hoarding toilet paper either,"

"I AM COMPLETELY IRATE at the way people throwing masks and gloves on the streets, in the parks, supermarket carts and parking lots."

"Seeing others who are not following ANY safety guidelines and acting confrontational toward those who do adhere to safety guidelines."

Seemingly unjust inter-personal tensions were reported as some of the main sources of difficulties, especially when they appeared to strain bonds of friendship and kinship:

"friends telling me I'm selfish for working!".

"My [child] stopped bringing my [grandchild] and is angry at me because I went back to work during the yellow Phase."

"I am being driven crazy by my family who have anxiety and nag me every time I go out. Their fears feel delusional and I refuse to validate them and this causes conflict."

Riots in the city made it difficult to cope, especially for those who were already impoverished and struggling with anxiety:

"I haven't applied for unemployment for mental hurdle reasons. I had started to settle in with the anxiety of it all, but in the past two days, with all the 'protests' happening, my anxiety levels have peaked high again. I believe this is because if history has proved anything we will be forced to stay inside longer due to their actions and I will lose further amounts of work."

Riots and government response to them (breakdown of law and order) have been reported by some as bigger source of stress than epidemic per se:

"The incessant fireworks from so many surrounding areas have created a MUCH tenser environment of PTSD for myself and famiky [sic] and neighbors [sic]. The constant shelling sounds, combined with the knowledge that Philly folks lots their minds and morality and attacked one another, storefronts, fires in the wake of Covid. . .THEN local politicians

abandoned, shamed and vilified our frontline defenders and protectors over a legal situation. . .more pressing and disheartening than Covid itself. Our civilization is not civil at all right now."

"People looting local shops, fear of police violence and escalation of social unrest during protests".

Respondents reported that their most difficult experiences arose from xenophobic abuse due to perception of Asian community as the source of contagion:

"The hate crimes against Asians Americans is extremely distressing and I am constantly terrified of what could happen to me or my friends or family. In addition, I am distress about the lack of visibility the Asian community has in the general public especially with a stigmatizing terms like the 'Chinese virus'."

One of the voiced sources of anxiety was not just loss of social interactions but a sense that there was a widespread social disintegration in the face of common threat:

"I knew there would be no post-9/11 feeling of unity before this even began. It's the division that gives me anxiety more than the virus."

**Coping.**   When the consequences of the epidemic became personal, it seemed harder for people to maintain equanimity. For example, death attributed to COVID-19 in their personal circle led to the sense of inevitability of infection, which produced their most difficult experience of the epidemic for some: "Death of people we know. . .. Closer and closer". On the other hand, some reported that the life under stay-at-home orders became easier as time went on, despite lingering fear of contagion, likely because early fears were not realized for these respondents:

"Initially I had great difficulty with the outbreak. I was not able to function normally for about 3 weeks. I have developed a fear of going outside and interacting with people and obsessive hand washing. I suffered with panic attacks, sleeplessness and cried multiple times a day. These things have lessened over the months since the stay at home [sic] order was issued. I still don't like interacting with people or going places."

Positive outlook appeared to be related to optimism about surviving the infection and acceptance of whatever may come:

"I enjoy more free time, more alone time, and more sleep. I'm not that worried about me or my family- vast majority who contract COVID19 survive. When are we not at risk of death?"

Although many people who were in poor health cited this as a source of most difficult experiences, some reported that their chronic mental health issues prepared them to cope better with added anxiety of the epidemic: ". . . it's like everyone else is joining the anxiety party, as if I've prepared for this moment all my life." People who already knew how to cope with mental health challenges reported being better off than those who experienced these for the first time due to the epidemic: ". . .this is easier for me than for many both due to my economic security and due to my previously developed coping strategies." Despite fear of infection because a

person has "an inclination to catching colds so . . . very cautious about . . . exposure", they "accept that the world and nature are reminding us who is really in charge" and "love the creative ways that people are finding to respond, change, do positive things", are "participating in some of that as we all try to find new ways to function in a world that is inevitably changing", and generally "find the quarantine to be a welcome change from a too-busy life".

Some people who were close with their families, with many generations living together appeared to fare rather well:

> "I live with my [child], her husband, and my grand [child]. Thus, I am totally supported by them and do not have the fears or concerns I would have if I lived alone or were head of a household.";

> "Having both [children] home from college has been helpful."

On the other hand, being house bound strained many close relationships: "I have . . . experienced somewhat more frequent verbal/emotional conflict with my spouse, which is both a product of stress and a cause of stress." Being sequestered indoors for unusually long time with family in close quarters was taking its toll, as with one respondent who was "tired of being with another person with zero alone time for months on end."

Parents struggled to explain the epidemic to children and why they cannot play with their friends, with the most difficult experience being "trying to get my child to understand why [they] can no longer play with . . . friends outside."

Some people reported that their "dogs really helped . . . isolation" and some reported getting a new puppy to help mitigate loneliness: "For someone living alone who lost their job, that changed everything for the better;" likewise, "getting a cat has been helpful."

Establishing a routine that was compatible to spending a lot of time indoors was helpful to some who "started watching Netflix at night", were "on the phone more and have spent considerable time doing a puzzle with . . . [grandchild] on Duo" were "thrilled to have many projects", "have begun to make yogurt and bread again after years", "have also been able to read more hours a day", and as the result reported to "have been relaxed mostly but miss touching family and friends."

Successful coping mechanisms that were reported included taking ownership on whatever freedom person had, creating a structure, and looking for support of family and friends despite existing limitations:

> "I am . . . not used to being home all of the time. . . . In the beginning one day blended into the next and I found myself feeling very depressed. I started making a schedule and I felt so much better. I highly recommend doing that and connecting with friends and family to keep your positivity in check! I will survive! 😊😊".

## Discussion

Our work should be treated as primarily descriptive in nature, even when we used tools commonly applied for statistical inference. Nonetheless, we demonstrated that two distinct, yet inter-related clusters of worries contributed to mood disorders in our sample: concern about hardships related to maintaining normalcy during epidemic plus fear of contagion. Furthermore, we believe that our quantitative analysis supports the notion that anxiety, depression and fear of infection increased as the first wave of COVID-19 epidemic progressed though Philadelphia. During the same period, the worries about hardships did not materially alter.

Both types of perceived risks (namely: food insecurity and subjective fear) were shown by Fitzpatrick et al. [6] to increase risk of depression during the epidemic in the US, after allowing for demographics, employment, heath, and social support. Quantitative and qualitative evidence that we examined converged to present a coherent picture of stressors that represented both the epidemic (threat of infection and severe illness) and measures taken by the state to combat the epidemic (economic hardships, restrictions on individual freedoms), as well as some of the means that residents of the city resorted to in finding support during this time.

To summarize more specifically, some difficulties and negative experiences appeared to be linked with increased anxiety and depression, which segregated into economic woes, general uncertainty of the situation, and health-related experiences. Access to food was clearly a major stressor among some participants. There was also a distinct group of participants reporting higher levels of worries who relied on government and social services for support. Fear of infection was linked to interaction with strangers and other health concerns. Having supportive "inner circle" of family, neighbours, and religious community appeared to be protective. Our findings largely agree with our in-depth look at sub-sample of our respondents who worked during the epidemic [9], within sub-sample who did not (S1 Table), and in related work among healthcare workers done using similar instruments at a roughly the same time in Philadelphia [10].

Examination of specific stories shared with us by the participants reveals a complex pattern of reactions and experiences during the first wave of the pandemic, not all negative. One overarching theme that appears to emerge is that respondents sought to control the situation and were blindsided by the stay-at-home orders that deprived them of agency in key decisions that affected their lives. This may have contributed to deterioration of mental health in the population because Fitzpatrick et al. [6] demonstrated that "mastery of fate" helped shield people during the pandemic from depression. Even among participants who saw merit in the government's actions, there was discontent about perceived unfairness of some of the measures and frustration with loss of freedom to decide what risks a person is willing to assume, e.g., to be with a dying relative. Some participants reported to enjoy slower pace of life resulting from the stay-at-home orders, which allowed them to spend more time with their families. We resist the temptation to add more layers to the themes we captured in our qualitative analysis, as we think that it is important for the participants to speak for themselves.

Our work suffers from numerous limitations. The principal among these is the selection mechanism by which participants enrolled in the study. They had to be healthy enough to participate and willing enough to disclose sensitive information in an online survey. Persons who were particularly afflicted with worries may have been more likely to try to respond to a long questionnaire, skewing our associations to be stronger than they are in the source population (selection bias). This may be evident in higher levels of anxiety among person who shared their stories with us, compared to those who did not. Thus, there is no chance that our sample is representative of the city of Philadelphia and there are not statistical techniques to compensate for this: sampling weights for key factors, such as mental health, are unknown. Our samples heavily skewed toward non-Hispanic Whites and likely did not adequately capture the experiences of racial and ethnic minorities. This is important to note because there is evidence of difference in rates of symptoms of mental ill health by race and ethnicity. For example, Fitzpatrick et al. [6], reported that non-Whites and those of Hispanic origin experienced higher rates of depressive symptoms compared to non-Hispanic Whites during the pandemic. Furthermore, since the pandemic, there was an increase of 150% in anti-Asian hate crime in the US and increase in hate incidents due to faulted stereotypes of Asian Americans and blame for the pandemic [24]. Documenting experiences of racial and ethnic minorities would require resources of community engaged research that was beyond our resources. If participants who

were more anxious and depressed were more likely to participate, we anticipated more of them to enrol early in the survey's timeline and lead to shallower than true slope in Fig 1 if there was a true increase in anxiety and depression during the pandemic (and associated effect estimates in regression given in Table 4). As some our participants told us, the timeline of the survey overlapped with social upheaval in the city, such that time-trend in anxiety and depression can be related to factors independent of the epidemic. Self-reported behaviours likely share correlated errors, again artificially manufacturing associations, but this is less likely to be at play with respect to demographics and is not a concern in examination of time-trends. The time-trends in anxiety, depression and hardships are in accord with time-trends in "deaths of despair" reported for the US as the whole by Mulligan [7] and is consistent with that estimated in early March 2020 for "distress" using individual-level data of representative sample of US population [2].

Among the strengths of our approach is the use of validated instruments to capture anxiety and depression, revealing rates that are elevated above norms in general population (median scores for anxiety in 5 to 6; for depression—around 3) when there are no external stressors at play [25]. This is in accord with published results by others [1–3]. We used rigorous methodology to validate that HADS worked as expected in our sample and derived two novel scores of specific worries correlated with mood disorder, which can be considered for monitoring during epidemics. We did manage to capture a large sample of individual experiences of residents of Philadelphia coping with COVID-19 epidemic in its early, most chaotic, stage. Even if not representative of the city, they do tell real stories representative of the participants, and suggest avenues for mitigation of negative unintended consequences of response to epidemics. The first (in no specific order) among these is to not add a threat of economic privation, food shortages and lack of medicine to an already volatile situation. The second is to focus on groups that are known to be vulnerable: those in poor health, younger, and less wealthy and ensure that their needs are addressed. Curiously, younger people (presumably with fewer health issues that put then at risk of COVID-19), with lower income, and with less education appear at elevated risk of mood disorders. This is like a report on 1,853 persons from Brazil [26] indicating that lower family income and younger ager were linked to anxiety and depression during the epidemic. The same study [26] reported that adhering to social distancing rules was linked with reduction in physical activity during pandemic, which in turn may have resulted in higher rates of mood disorders. This likewise resonates with our observation of higher levels of anxiety and depression among those who went outside less frequently, although we cannot exclude reverse causation due to lack of pre-pandemic information in our data. It seems that creating conditions that discourage physical activity when mental health is at threat is ill-advised. We are willing to hazard a guess that addressing these needs would require knowledge about people concerned, best obtained proactively during preparation of hypothetical epidemic, not during it.jj.

Our data implies (does not demonstrate) that, to the extent possible, it appears sensible to empower families, friends, and neighbours to look after each other: that is where most people naturally sought support in turbulent times and were better off when they could report that they were sure that such support will be available. Because participants who reported that they can count on support from their families, neighbours and religious communities were less anxious and depressed, it appears sensible to consider whether epidemic-mitigation measures undermine the availability of these forms of support and seek to, at the very least, minimize disruptions of these support networks. Persons who were single were at increased risk (just as for depression in [6]), supporting the notion that immediate family can be helpful, but not those who were widowed or divorced, perhaps due to higher resilience that comes with older age, a survivor effect. Corroborating our observations, the stronger personal ties were

protective against depression in survey of Fitzpatrick et al. [6]. Reliance on government and social organizations appeared to be related to elevated risk but we do not know just how high the risk was before support was sought and whether it was sought at all. The missing historical context is important because those relying on government and social services are likely to be vulnerable, either economically or in terms of mental health (or both), even in absence of the epidemic. However, examination of detailed narratives indicates that some respondents did feel let down by government and social organizations. Some unpleasant experience that may be resulted in stigmatization (being upset by others avoiding one in public) can be yet another unintended consequence of the push for "social distancing". It is hard to judge people for acting on their fears, but it seems that kindness to strangers and not instilling fear of strangers in the public may have been helpful. From this perspective, evaluation of media and public health messaging in the relevant issue may reveal insights of plausible preventable causes of such hurtful events.

It is important to note that our data indicates that participants took sensible precautions that were prescribed and that the extent to which they complied was related to their anxieties. It is not appropriate to draw causal inference from such correlations and we hope that our findings are not taken to mean that there is benefit in inducing anxiety for the sake of compliance with public health measures. Surely there are more ethical ways to convince people to change their behaviour, on which a voluminous behavioural science literature exists. This is important to reflect on because both our findings and those of Fitzpatrick et al. [6] indicate that fear of infection increases risk of depression.

We conclude our description of how some residents of Philadelphia coped with the first wave of COVID-19 epidemic with the expression of hope that experiences that we helped to bring to light through application of rigorous research methods will help public health authorities and governments to lessen suffering that people endure during current and forthcoming epidemics, and similar events. We also suspect that sharing stories of our participants with wider readership will have the benefit of validating experiences that some people so far have kept private. For that opportunity, we are most grateful to all those who contributed to our work.

## Supporting information

**S1 Table. Demographics of participants who (a) did not report employment at start of the epidemic and (b) those who shared narratives on the most difficult experiences (see Table 1 for definitions of abbreviations).**
(PDF)

**S2 Table. Precautions reported during survey period.**
(PDF)

**S3 Table. Correlations of support, worries and themes of most difficult experiences with anxiety and depression scores for those who did not report employment at the start of epidemic see Table 2 for definitions of abbreviations).**
(PDF)

**S4 Table. Details of factor analysis in support of Fig 2 and Table 3.**
(PDF)

**S5 Table. Interaction of coded themes of most difficult experiences with gender on anxiety, depression, hardships, and fear of infection; see Table 4 and text for details of models to**

**which cross-product interaction terms were added.**
(PDF)

## Acknowledgments

The authors are deeply indebted to all the participants who responded to survey while learning to live under extreme pressures precipitated by the COVID-19 pandemic. Dr Nicola M. Cherry of the University of Alberta generously shared ideas and materials on related research. We wish to thank Todd Wolfson and Briar Smith for their feedback on the survey instrument, Mariela Morales for her assistance with online advertisement and Spanish translation, Guangzi Song and Xi Wang for help with the Chinese translation, and Duong (Tina) Nguyen for help with the Vietnamese translation.

## Author Contributions

**Conceptualization:** Igor Burstyn, Tran B. Huynh.

**Data curation:** Igor Burstyn, Tran B. Huynh.

**Formal analysis:** Igor Burstyn, Tran B. Huynh.

**Investigation:** Igor Burstyn, Tran B. Huynh.

**Methodology:** Igor Burstyn, Tran B. Huynh.

**Project administration:** Igor Burstyn, Tran B. Huynh.

**Writing – original draft:** Igor Burstyn, Tran B. Huynh.

**Writing – review & editing:** Igor Burstyn, Tran B. Huynh.

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
