## [Decision Letter · Decision Letter 0]

23 Aug 2021

PONE-D-21-11341

Experiences of coping with the first wave of COVID-19 epidemic in Philadelphia, PA: Mixed methods analysis of a cross-sectional survey of worries and symptoms of mood disorders.

PLOS ONE

Dear Dr. Burstyn,

Thank you for submitting your manuscript to PLOS ONE. After careful consideration, we feel that it has merit but does not fully meet PLOS ONE’s publication criteria as it currently stands. Therefore, we invite you to submit a revised version of the manuscript that addresses the points raised during the review process.

We look forward to receiving your revised manuscript.

Kind regards,

Vijayaprasad Gopichandran

Academic Editor

PLOS ONE

Journal Requirements:

Furthermore, please note that all PLOS journals ask authors to adhere to our policies for sharing of data and materials: https://journals.plos.org/plosone/s/data-availability. According to PLOS ONE’s Data Availability policy, we require that the minimal dataset underlying results reported in the submission must be made immediately and freely available at the time of publication. As such, please remove any instances of 'unpublished data' or 'data not shown' in your manuscript and replace these with either the relevant data (in the form of additional figures, tables or descriptive text, as appropriate), a citation to where the data can be found, or remove altogether any statements supported by data not presented in the manuscript.

4. Please provide additional information regarding the eligibility criteria for the study. In particular please clarify whether your study included any minors and If your study included minors under age 18, state whether you obtained consent from parents or guardians. If the need for consent was waived by the ethics committee, please include this information.

6. Please upload a new copy of Figure 1 as the detail is not clear. Please follow the link for more information: https://blogs.plos.org/plos/2019/06/looking-good-tips-for-creating-your-plos-figures-graphics/" https://blogs.plos.org/plos/2019/06/looking-good-tips-for-creating-your-plos-figures-graphics/.

Reviewers' comments:

Reviewer's Responses to Questions

**Comments to the Author**

1. Is the manuscript technically sound, and do the data support the conclusions?

Reviewer #1: Yes

Reviewer #2: Yes

Reviewer #3: Partly

2. Has the statistical analysis been performed appropriately and rigorously? 

Reviewer #1: Yes

Reviewer #2: Yes

Reviewer #3: Yes

3. Have the authors made all data underlying the findings in their manuscript fully available?

Reviewer #1: No

Reviewer #2: No

Reviewer #3: No

4. Is the manuscript presented in an intelligible fashion and written in standard English?

Reviewer #1: Yes

Reviewer #2: Yes

Reviewer #3: Yes

5. Review Comments to the Author

Reviewer #1: The paper is well presented; the question is valid and of general interest.

A few additional points I think would improve the paper are:

- A reference supporting the use of HADS in the general population, as well as the use of the specified cutoffs for significant anxiety and depression.

- A description of the sub sample that provided qualitative data

- A review of grammar and spelling (for eg: the word 'rely' seems to have been replaced with 'reply' through the manuscript)

Reviewer #2: 1. The manuscript is technically sound, with elaborate explanations on study tools used and the methods of analysis performed. The positive points include use of a validated tool (HADS) and also a deductive approach based on lived experiences in qualitative part to frame the themes.

2. Statistical analysis is explained elaborately. However, there are certain points to consider for revision.

The normality of the data is not mentioned in the manuscript. This seemed pertinent as the tables present the mean scores (Table 1) while the figure (Figure 1) presents the median and interquartile range.

Though the missing values are imputed, it is seen that they are only mentioned as a separate entity in the table.

Use of rank biserial correlation is appreciable, considering the categorisation of HADS as a binomial outcome. However, since HADS and the other outcomes estimated (Sources of support, worries about COVID-19, “most difficult experiences”) are rated on a continuous Likert scale of 0-100, it makes me think why not Pearson correlation co-efficient, especially when these values of correlates and statistical significance also indicate the univariate regression analysis.

Maximum likelihood factor analysis is not delineated.

In table 3, the standardised scoring co-efficients which have a significant factor loading are emboldened (as stated in the manuscript), but the last variable is also mentioned to be non-significant.

Oblique VARIMAX rotation is mentioned in the statistical part, but providing a picture of it or a detailed explanation could have been better. Also, the scree plot.

In the multiple regression analysis, interaction of the gender is explained in the text and also in the description of statistical analysis, but not in the table.

3. The overall writing of the manuscript is cohesive and comprehensible. There are places at which the language could be improved. There are many places at which words are misspelt, lack of prepositions, wrong usage of tenses, grammatical errors.

Reviewer #3: Refrain from using language establishing causal relationships between mental state and experience during COVID from data of this cross sectional study. Concluding the study with clarification about such statements is not sufficient as the specific results can be further cited by other researchers. For the same, authors have to be cautious in writing results and giving misinterpretation of research outcomes.

6. PLOS authors have the option to publish the peer review history of their article (what does this mean?). If published, this will include your full peer review and any attached files.

Reviewer #1: **Yes: **Dr. Abhinav Chichra

Reviewer #2: No

Reviewer #3: No

---

## [Author Response · Author response to Decision Letter 0]

16 Sep 2021

this was attached as a separate PDF file

---

## [Editor Report · Decision Letter 1]

22 Sep 2021

Experiences of coping with the first wave of COVID-19 epidemic in Philadelphia, PA: Mixed methods analysis of a cross-sectional survey of worries and symptoms of mood disorders.

PONE-D-21-11341R1

Dear Dr. Burstyn,

We’re pleased to inform you that your manuscript has been judged scientifically suitable for publication and will be formally accepted for publication once it meets all outstanding technical requirements.

Kind regards,

Vijayaprasad Gopichandran

Academic Editor

PLOS ONE
---

## [Editor Report · Acceptance letter]

23 Sep 2021

PONE-D-21-11341R1 

Experiences of coping with the first wave of COVID-19 epidemic in Philadelphia, PA: Mixed methods analysis of a cross-sectional survey of worries and symptoms of mood disorders. 

Dear Dr. Burstyn:

I'm pleased to inform you that your manuscript has been deemed suitable for publication in PLOS ONE. Congratulations! Your manuscript is now with our production department. 

Kind regards, 

on behalf of

Dr. Vijayaprasad Gopichandran 

Academic Editor

PLOS ONE